# Ginseng Extracts, GS-KG9 and GS-E3D, Prevent Blood–Brain Barrier Disruption and Thereby Inhibit Apoptotic Cell Death of Hippocampal Neurons in Streptozotocin-Induced Diabetic Rats

**DOI:** 10.3390/nu12082383

**Published:** 2020-08-09

**Authors:** Jee Youn Lee, Chan Sol Park, Hae Young Choi, Tae Young Yune

**Affiliations:** 1Age-Related and Brain Diseases Research Center, Kyung Hee University, Seoul 02447, Korea; jeeyoun@khu.ac.kr (J.Y.L.); chansol1028@khu.ac.kr (C.S.P.); neuron@khu.ac.kr (H.Y.C.); 2Department of Biomedical Science, Kyung Hee University, Seoul 02447, Korea; 3Department of Biochemistry and Molecular Biology, School of Medicine, Kyung Hee University, Seoul 02447, Korea; 4KHU-KIST Department of Converging Science and Technology, Kyung Hee University, Seoul 02447, Korea

**Keywords:** blood–brain barrier, apoptosis, hippocampus, microglia, astrocyte, streptozotocin

## Abstract

Type 1 diabetes mellitus is known to be linked to the impairment of blood–brain barrier (BBB) integrity following neuronal cell death. Here, we investigated whether GS-KG9 and GS-E3D, bioactive ginseng extracts from Korean ginseng (*Panax ginseng* Meyer), inhibit BBB disruption following neuronal death in the hippocampus in streptozotocin-induced diabetic rats showing type 1-like diabetes mellitus. GS-KG9 and GS-E3D (50, 150, or 300 mg/kg, twice a day for 4 weeks) administered orally showed antihyperglycemic activity in a dose-dependent manner and significantly attenuated the increase in BBB permeability and loss of tight junction proteins. GS-KG9 and GS-E3D also inhibited the expression and activation of matrix metalloproteinase-9 and the infiltration of macrophages into the brain parenchyma, especially into the hippocampal region. In addition, microglia and astrocyte activation in the hippocampus and the expression of proinflammatory mediators such as *tnf-α*, *Il-1β*, *IL-6*, *cox-2*, and *inos* were markedly alleviated in GS-KG9 and GS-E3D-treated group. Furthermore, apoptotic cell death of hippocampal neurons, especially in CA1 region, was significantly reduced in GS-KG9 and GS-E3D-treated groups as compared to vehicle control. These results suggest that GS-KG9 and GS-E3D effectively prevent apoptotic cell death of hippocampal neurons by inhibiting BBB disruption and may be a potential therapy for the treatment of diabetic patients.

## 1. Introduction

Diabetes mellitus, characterized by chronic hyperglycemia, is a metabolic disorder with semeiotic symptoms such as polyuria, polyphagia, and polydipsia along with low-grade inflammation [1]. Recent clinical and experimental studies suggest that prolonged hyperglycemic conditions influence a progressive impairment of neuronal function in the brain [2,3,4]. Diabetes mellitus has also been associated with increased severity of epileptic seizure [5] and risk of mortality following traumatic brain injury [6]. In particular, diabetes mellitus has been strongly linked to mild cognitive impairments [7,8] and is considered a predisposing factor for developing vascular dementia [9] and Alzheimer’s disease [10]. Learning and memory are also known to be affected, especially in adults with poor glycemic control, longer duration of diabetes, or young age of onset [11,12].

Altered glycemic conditions observed in diabetic patients are prodromal to blood–brain barrier (BBB) impairment [13,14,15,16]. Experimental evidence from in vitro and in vivo studies has shown that BBB integrity in diabetes is somewhat compromised, resulting in increased barrier permeability [13,14,17,18]. High glucose-induced oxidative stress leads to brain tissue damage, resulting in increased BBB permeability associated with cerebral dysfunction. Inflammatory factors are also involved in the neuron degeneration, apoptosis, and even necrosis caused by the destruction of BBB integrity and increased permeability [19,20,21,22]. The tight junction is the major component that maintains the structural and functional integrity of the BBB. Occludin, claudin-5, and zonula occludens (ZO)−1 are important tight junction-associated proteins that sustain the permeability for low-molecular-mass molecules as well as the electrical resistance of brain barrier tissue. Thus, the loss of occludin, claudin-5, and ZO-1 proteins or their translocation might be connected with increased BBB permeability [23,24].

Red or white ginseng and its extracts have been used as traditional medicines and functional foods in many countries. Studies have examined the effects of the resulting ginsenosides in various disease models such as metabolic disorders [25,26,27], cancer [28,29], immune response [30,31], skin care [32,33], pulmonary system [34], brain injury [35,36], depression [37], and Alzheimer’s disease in both in vitro and in vivo models [38,39]. Here, we examined the effect of GS-E3D, pectin lyase-modified red ginseng extracts, and GS-KG9, an air-dried and ethanol extracted white ginseng extract, on BBB disruption following hippocampal neuron cell death in experimental streptozotocin (STZ)-induced diabetic rats.

## 2. Materials & Methods

### 2.1. Preparation of GS-KG9 and GS-E3D

Korean ginseng (*Panax ginseng* Meyer, 4-year-old) was purchased from a local market (Geumsan, Korea). GS-KG9 was prepared according to a previous report [40]. Briefly, a mixture of white ginseng and white ginseng tail (6:4, *w*/*w*) dried at 55 °C for 5 d in an air dryer was extracted with 70% ethanol three times at 40 °C. The white ginseng extract was filtered and concentrated in vacuo and named GS-KG9. The dried GS-KG9 included 116.3 mg/g crude saponin, as previously described [40].

GS-E3D was prepared as described in a previous report [41]. Briefly, red ginseng extract, adjusted to 6 Brix, was incubated with 10% pectin lyase (EC 4.2.2.10; Novozyme, #33095 Bagsværd, Denmark) at 50 °C for 5 d in a shaking incubator (150 rpm). To terminate the reaction, the processed extract was incubated at 95 °C for 10 min and then freeze-dried for storage prior to the subsequent experiments. The dried GS-E3D consisted of 120.2 mg/g crude saponin, as previously described [41].

**Animal model and drug administration.** Male SD injection of S TZ (60 mg/kg Sigma, St. Louis, MO, USA) and control animals received a same volume of saline [40]. Four days after STZ injection, hyperglycemia was confirmed by measuring fasting blood glucose level in a tail vein using an Accu-Chek Compact Plus glucose meter (Roche Diagnostics, Meylan, France). Animals showing blood glucose levels over 250 mg/dL were selected and randomly divided into each experimental group including STZ control, GS-KG9, and GS-E3D. GS-KG9 or GS-E3D (50, 150, or 300 mg/kg) was dissolved in saline and administrated orally twice a day for 4 weeks. STZ control group received equivolumetric oral administration of saline at the corresponding time points. For normal controls, rats received no pharmacological treatment. Body weight and water (mL/day) and food (g/day) consumption were determined every day during the 4-week experimental period. All experimental procedures involving animals complied with the commonly accepted “3Rs” and the ARRIVE guidelines (Animal Research: Reporting of in vivo Experiments). Our experimental protocols were approved by the Institutional Animal Care and Use Committee (IACUC) of Kyung Hee University (permission number: KHUASP(SE)-15-069).

### 2.2. Measurement of Blood–Brain Barrier Disruption

The integrity of the BBB was investigated with Evans blue dye extravasation according to previous reports [42,43], with a few modifications. At 28 d after starting GS-KG9 or GS-E3D treatment, 5 mL of 2% Evans blue dye (Sigma) solution in saline was administered i.p. Three hours later, the animals were perfused with PBS (Phosphate buffered saline) and the brain hippocampus region was removed and homogenized in a 50% trichloroacetic acid solution. After homogenization, samples were centrifuged at 10,000× *g* for 10 min, supernatants were collected and fluorescence was quantified at an excitation wavelength of 620 nm and an emission wavelength of 680 nm. The amount of dye in samples was determined as micrograms per gram of tissue from a standard curve plotted using known amounts of dye. For qualitative analysis, some animals were perfused with PBS and subsequently with 4% formaldehyde, as described above. The brains were then immersed in a 30% sucrose solution and brain tissue were cut into 20 μm thick sections with a cryostat. The fluorescence of Evans blue in the hippocampus was observed with a fluorescence microscope and the relative fluorescence intensity was determined by MetaMorph software (Molecular devices, Sunnyvale, CA, USA).

**Tissue preparation.** Rats were anesthetized with chloral hydrate (500 mg/kg, intraperitoneal injection) and perfused transcardially with phosphate buffer (100 mM, pH 7.4) followed by ice-cold 4% paraformaldehyde, they were and then decapitated. The brains were removed and post-fixed overnight in phosphate buffer (50 mM, pH 7.4) containing 4% paraformaldehyde. The brains were then immersed in a 30% sucrose solution (in 50 mM phosphate-buffered saline, PBS) and stored at 4 °C until sectioning. Frozen brains were sectioned along the coronal plane (30 μm) using a cryostat (Leica Microsystems AG, Wetzlar, Germany) and maintained in a storage solution at 4 °C. For molecular work, rats were perfused with PBS and bilateral hippocampal tissue samples were isolated and frozen at −80 °C.

### 2.3. Immunohistochemistry

Frozen sections were processed for immunohistochemistry with antibodies against OX-42 (1:100, Merck Millipore, Billerica, MA, USA) for staining microglia, GFAP (Glial fibrillary acidic protein) (1:5000, Merck Millipore) for staining astrocytes, and ED-1 (Bio-Rad, Hercules, CA, USA) for staining macrophages, as previously described [42]. The sections were incubated with primary antibodies, followed by biotin-conjugated secondary antibodies (Dako, Carpinteria, CA, USA). The ABC (Avidin-biotin complex)method was used to detect labeled cells using a Vectastain kit (Vector Labs, Burlingame, CA, USA). DAB (3, 3-diaminobenzidine)served as the substrate for peroxidase. For immunofluoresence staining, cy3-conjugated secondary antibodies (Jackson ImmunoResearch, West Grove, PA, USA) were used. In addition, nuclei were labeled with DAPI (4′,6-diamidino-2-phenylindole)according to the protocol of the manufacturer (ThermoFisher Scientific, Waltham, MA, USA). In all controls, reaction to the substrate was absent if the primary antibody was omitted or if the primary antibody was replaced by a non-immune, control antibody. Serial sections were also stained for histological analysis with Cresyl violet acetate. The counting of ED-1-positive cells was carried out by the investigators, who were blind to the experimental conditions. Every five sections throughout the hippocampus were selected and the ED-1-positive cells in the CA1 area (10 sections per animal) were counted and averaged.

### 2.4. Nissl Staining

After being mounted onto gelatin-coated slides, tissue sections were stained with 0.5% Cresyl violet, dehydrated through graded alcohols (70%, 80%, 90%, and 100% × 2), placed in xylene, and covered with a coverslip after the addition of permount. The Nissl-stained cells in the CA1 area were counted by a researcher who was blinded to the experimental conditions. Every five sections throughout the hippocampus were processed for counting (10 sections per animal). The number of cells in CA1 was quantitatively expressed as a percentage compared to the normal control.

### 2.5. TUNEL Staining

To examine apoptotic cell death of hippocampal neurons, coronal sections including hippocampal regions were processed for terminal deoxynucleotidyl transferase-mediated deoxyuridine triphosphate-biotin nick end labeling (TUNEL) staining using an Apoptag in situ kit (Merck Millipore). Investigators who were blind to the experimental conditions carried out all TUNEL analyses. Every five sections throughout the hippocampus were selected and the TUNEL-positive cells in the CA1 area (10 sections per animal) were counted and averaged.

### 2.6. Western Blot

Total protein isolated from hippocampal tissue at 28 d after starting the drug treatment was prepared with a lysis buffer containing 50 mM Tris-HCl, pH 8.0, 150 mM NaCl, 1% NP-40, 0.5% deoxycholate, 0.1% SDS, 10 mM Na_2_P_2_O_7_, 10 mM NaF, 1 μg/mL aprotinin, 10 μg/mL leupeptin, 1 mM sodium vanadate, and 1 mM PMSF (Phenylmethylsulfonyl fluoride). Tissue homogenates were incubated for 20 min at 4 °C and centrifuged at 25,000× *g* for 30 min at 4 °C. The protein concentration was determined using the BCA assay kit (ThermoFisher Scientific, Rockford, IL, USA). Protein sample (40 μg) was separated on SDS-PAGE and transferred to nitrocellulose membrane (Merck Millipore). The membranes were blocked in 5% nonfat skim milk or 5% bovine serum albumin in TBST (Tris-buffered saline, 0.1% Tween-20)for 1 h at room temperature, followed by incubation with antibodies against occludin (1:1000, ThermoFisher Scientific), ZO-1 (1:1000, ThermoFisher Scientific), and claudin-5 (1:1000, ThermoFisher Scientific). The primary antibodies were detected with horseradish peroxidase-conjugated secondary antibodies (Jackson ImmunoResearch). Immunoreactive bands were visualized by chemiluminescence using Supersignal (ThermoFisher Scientific). β-tubulin (1:20,000; Sigma) was used as an internal control. Experiments were repeated three times and the densitometric values of the bands on Western blots obtained by AlphaImager software (Alpha Innotech Corporation, San Leandro, CA, USA) were subjected to statistical analysis. The backgrounds of the films were subtracted from the optical density measurements.

### 2.7. Gelatin Zymography

The activity of matrix metalloproteinase (MMP)-2 and -9 at 28 d after starting the drug treatment was examined by gelatin zymography based on a previously described protocol, with some modifications [42]. Briefly, hippocampal tissue samples were weighed and homogenized in lysis buffer containing the following: 28 mM Tris-HCl, 22 mM Tris-base, pH 8.0, 150 mM NaCl, 1% Nonidet P-40, 0.5% sodium deoxycholate, and 0.1% SDS (Sodium dodecylsulfate-polyacrylamide). The protein concentration of the homogenates was determined by the bicinchoninic acid method (BCA protein assay kit, Pierce, Rockford, IL, USA). After determination of protein concentration of the homogenates, equal amounts of protein (30 µg) were loaded on a Novex 10% zymogram gel (ThermoFisher Scientific) and separated by electrophoresis with 100 V (19 mA) at 4 °C for 6 h. The gel was then incubated with renaturing buffer (2.5% Triton X-100) at room temperature for 30 min to restore the gelatinolytic activity of the proteins. After incubation with developing buffer (50 mM Tris-HCl, pH 8.5, 0.2 M NaCl, 5 mM CaCl_2_, 0.02% Brii35) at 37 °C for 24 h, the gel was stained with 0.5% Coomassie blue for 60 min and then destained with 40% methanol containing 10% acetic acid until appropriate color contrast was achieved. Clear bands on the zymogram were indicative of gelatinase activity. Relative intensity of zymography (relative to sham or vehicle) was measured and analyzed by AlphaImager software (Alpha Innotech Corporation). Backgrounds were subtracted from the optical density measurements.

### 2.8. RNA Isolation and RT-PCR

Total RNA was isolated using TRIZOL Reagent (ThermoFisher Scientific) and 0.5 μg of total RNA was reverse-transcribed into first-strand cDNA using MMLV (Moloney Murine Leukemia Virus), according to the manufacturer’s instructions (ThermoFisher Scientific). For PCR amplifications, the following reagents were added to 1 μL of first-strand cDNA: 0.5 U taq polymerase (Takara, Kyoto, Japan), 20 mM Tris-HCl, pH 7.9, 100 mM KCl, 1.5 mM MgCl_2_, 250 μM dNTP, and 10 pmole of each specific primer. The primers used for *mmp-2*, *mmp-9*, *tnf-α*, *il-1β*, *il-6*, *cox-2*, *inos*, and *gapdh* were synthesized by the Genotech (Daejeon, Korea) and the sequences of the primers are as follows (5′-3′): *mmp-2* forward, 5′-ACC ATC GCC CAT CAT CAA GT-3′, reverse, 5′-CGA GCA AAA GCA TCA TCC AC-3′; *mmp-9* forward, 5′-AAA GGT CGC TCG GAT GGT TA-3′, reverse, 5′-AGG ATT GTC TAC TGG AGT CGA-3′; *tnf-α* forward, 5′-CCC AGA CCC TCA CAC TCA GAT-3′; reverse, 5′-TTG TCC CTT GAA GAG AAC CTG-3′; *il-1β* forward, 5′-GCA GCT ACC TAT GTC TTG CCC GTG-3′, reverse, 5′-GTC GTT GCT TGT CTC TCC TTG TA-3′; *il-6* forward, 5′-AAG TTT CTC TCC GCA AGA TAC TTC CAG CCA-3′; reverse, 5′-AGG CAA ATT TCC TGG TTA TAT CCA GTT-3′; *cox-2* forward, 5′-CCA TGT CAA AAC CGT GGT GAA TG-3′; reverse, 5′-ATG GGA GTT GGG CAG TCA TCA G-3′; *inos* forward, 5′-CTC CAT GAC TCT CAG CAC AGA G-3′; reverse, 5′-GCA CCG AAG ATA TCC TCA TGA T-3′; *gapdh* forward, 5′-AAC TTT GGC ATT GTG GAA GG-3′; reverse, 5′-GGA GAC AAC CTG GTC CTC AG-3′. The plateau phase of the PCR reaction was not reached under these PCR conditions. After amplification, PCR products were subjected to 1.5–2% agarose gel electrophoresis and visualized by ethidium bromide staining. The relative density of bands (relative to sham value) was analyzed by the AlphaImager software (Alpha Innotech Corporation). Experiments were repeated three times and the values obtained for the relative intensity were subjected to statistical analysis. The gels shown in figures are representative of results from three separate experiments.

### 2.9. Statistical Analysis

All data are presented as the mean ± standard deviation (SD). Comparisons between STZ control and GS-KG9 or GS-E3D-treated groups were made by unpaired Student *t* test. Multiple comparisons between groups were performed by one-way ANOVA. Tukey’s multiple comparison was used for post-hoc analysis. Statistical significance was accepted with *p* < 0.05. Statistical analyses were performed using SPSS 15.0 (SPSS Science, Chicago, IL, USA).

## 3. Results and Discussion

### 3.1. GS-KG9 and GS-E3D Inhibit Hyperglycemia in STZ-Induced Diabetic Rats

To induce hyperglycemia, we injected STZ (60 mg/kg) into rats intraperitoneally (i.p.), which resulted in diabetic syndromes verified by the presence of polydipsia, polyuria, hyperglycemia, and weight loss in the diabetic animals [44]. Hyperglycemia was confirmed by measuring the blood glucose level in a tail vein and then the animals showing blood glucose levels over 250 mg/dL were selected and randomly divided into each experimental group (Figure 1). At 4 days after STZ injection, the mean blood glucose levels were significantly higher in STZ-induced diabetic rats than in normal rats (STZ, 364.4 ± 24.5 mg/dL vs. normal, 112.0 ± 5.5 mg/dL). We first determined whether GS-KG9 and GS-E3D reduce glucose level in STZ-induced diabetic rats. As shown in Figure 2A, in STZ-injected diabetic rats, the blood glucose level was increased and maintained at a high level during the experiments (7 d, 337.3 ± 55.2; 14 d, 324.3 ± 32.2; 21 d, 345.9 ± 39.9; 28 d, 342.4 ± 37.8 mg/dL). However, both GS-KG9 and GS-E3D (150 and 300 mg/kg) significantly reduced the level of blood glucose at 28 d after drug treatment in a dose-dependent manner. Our results also showed that a dose of 300 mg/kg of the drugs was an optimal dose to achieve an anti-hyperglycemic effect in STZ-induced diabetic rats (Figure 2A) and thus this dosage was used in these experiments. Especially, GS-KG9 or GS-E3D (300 mg/kg) treatment significantly decreased the blood glucose level from 14 d after drug treatment as compared to the STZ control group (at 28 d: STZ, 342.4 ± 37.8; STZ+GS-KG9, 240.4 ± 32.1; STZ+GS-E3D, 238.3 ± 43.0 mg/dL) (Figure 2B). Evident body weight loss (Figure 2C), polyphagia (Figure 2D), and polydipsia (Figure 2E) were observed in STZ-induced diabetic rats, whereas both body weight loss and polydipsia in STZ-treated rats were significantly alleviated by GS-E3D treatment from 20 d and 28 d after administration, respectively (Figure 2C,D). Meanwhile, the body weight and polydipsia in GS-KG9-treated groups were slightly higher than in the STZ-treated control group, although not to a level of significance (Figure 2C,D). In addition, polyphagia was not significantly changed by GS-KG9 or GS-E3D treatment (Figure 2E). In addition, any significant change in body weight and toxicity signs (such as piloerection, alteration in the locomotor activity, or diarrhea) was not observed in either of the experimental groups or normal rats during the experiment (data not shown). These results indicate that GS-KG9 and GS-E3D reduce glucose levels and improve hyperglycemia-associated symptoms in STZ-induced diabetic rats.

It is known that insulin causes a reduction in blood glucose by facilitating the uptake and storage of glucose. In addition, the blood glucose level reduction can also depend on a downregulation of carbohydrate hydrolyzing enzymes such as α-amylase and α-glucosidase in digestive tract. α-amylase is the enzyme that hydrolyzes starch to maltose and consequentially higher postprandial hyperglycemia, while α-glucosidase is responsible for the breakdown of oligo- and/or disaccharides to monosaccharides [45]. The inhibition of these enzymes leads to a decrease in blood glucose level, because monosaccharides are a form of carbohydrate which is absorbed through the mucosal border in the small intestine [46]. It has been demonstrated that Panax ginseng extracts ameliorate hyperglycemia in STZ-induced diabetic animal models by increasing serum insulin levels [47,48]. In addition, a recent report showed that Korean red ginseng extract inhibits α-glucosidase and α-amylase activities and decreases glucose uptake and transport rate in human colon cell lines [48]. Although the level of insulin and the activity of carbohydrate hydrolyzing enzymes was not examined in the present study, we cannot rule out the possibility that GS-KG9 and GS-E3D may be involved in the downregulation of serum insulin levels or inhibition of glucose production and glucose absorption in the digestive tract, thereby reducing the blood glucose level in STZ-induced diabetic rats. We will seek to elucidate the precise mechanism underlying GS-KG9 and GS-E3D-mediated inhibition of blood glucose level in a future study.

### 3.2. GS-KG9 and GS-E3D Inhibit the Increase in BBB Permeability in the Hippocampal Regions of STZ-Induced Diabetic Rats

It is known that all forms of diabetes are characterized by chronic hyperglycemia, resulting in the development of a number of microvascular and macrovascular pathologies. Diabetes is also associated with changes in brain microvasculature, leading to dysfunction and ultimately BBB disruption [17,49,50,51]. Since GS-KG9 and GS-E3D caused a significant reduction in blood glucose levels in STZ-induced diabetic rats (in Figure 2), we next investigated whether STZ-induced hyperglycemia would lead to microvascular changes and whether GS-KG9 and GS-E3D inhibit BBB disruption in the brain parenchyma, especially in the hippocampal region. To examine the change in BBB permeability, we injected Evans blue dye into the STZ-treated rats (i.p.) administered for four weeks with GS-KG9 and GS-E3D. As a result, we found leakage of Evans blue dye into the brain parenchyma of STZ-induced diabetic rats, especially into the hippocampal region, but not in normal rats, which means that STZ-induced hyperglycemia resulted in BBB disruption. Furthermore, GS-KG9 or GS-E3D treatment significantly reduced the amount of Evans blue dye extravasation when compared with the STZ control (STZ, 20.3 ± 2.7; STZ+GS-KG9, 9.8 ± 2.1; STZ+GS-E3D, 7.3 ± 2.3 mg/dL) (Figure 3A). Consistent with these results, the fluorescence microscopy images of hippocampal CA1 regions of STZ-induced diabetic rat brains showed the presence of Evans blue dye (bright red) in the brain parenchyma around blood vessels, demonstrating the leakage of the dye across the BBB. However, the intensity of Evans blue was reduced in GS-KG9 or GS-E3D-treated groups (Figure 3B,C). These results indicate that GS-KG9 and GS-E3D inhibit BBB disruption by STZ-induced hyperglycemia.

### 3.3. GS-KG9 and GS-E3D Inhibit the Expression and Activity of MMP-9 in STZ-Induced Diabetic Rats

It has been known that hyperglycemia increases matrix metalloproteinase (MMP) activity and thereby results in the breakdown of the tight junction following the increase in BBB permeability [14]. MMPs are a family of zinc-dependent proteolytic enzymes that degrade components of the extracellular matrix in various pathophysiological conditions. The excessive proteolytic activity of MMPs such as MMP-2 and MMP-9 results in BBB disruption after CNS injury such as stroke and spinal cord injury [42,52,53]. Recently, Aggarwal et al. [54] reported that MMP-9 activity is significantly elevated in STZ-induced diabetic rat brains and the inhibition of MMP-9 activity leads to the restoration of BBB integrity and improves learning and memory in STZ-induced diabetic rats. Since GS-KG9 and GS-E3D treatment reduced BBB disruption in the hippocampus of STZ-induced diabetic rats, we examined whether these drugs would inhibit the expression and activity of MMP-2 and/or MMP-9. As shown in Figure 4A, the levels of *mmp-9* mRNA increased in STZ-injected rats compared with normal rats. Furthermore, *mmp-9* mRNA expression was significantly inhibited by GS-KG9 or GS-E3D administration compared with STZ control group (Figure 4A,B). Using gelatin zymography, GS-KG9 and GS-E3D also significantly inhibited the increase in MMP-9 activity (active MMP-9 band) as compared with the STZ control (Figure 4C,D) (active MMP-9; STZ, 8.7 ± 0.6; GS-KG9, 6.1 ± 0.7; GS-KG9, 6.1 ± 0.7). No significant change was observed in either mRNA expression or gelatinase activity of MMP-2 in diabetic rats, as reported [54]. Our data thus indicate that GS-KG9 and GS-E3D reduce both *mmp-9* mRNA expression and MMP-9 activation in chronic hyperglycemic diabetic rats.

### 3.4. GS-KG9 and GS-E3D Alleviate the Loss of Tight Junction Proteins in the Hippocampus of STZ-Induced Diabetic Rats

Since we showed that GS-KG9 and GS-E3D inhibited BBB disruption in the hippocampus of STZ-induced diabetic rats (Figure 3), we next examined whether these drugs would also inhibit the loss of tight junction proteins by Western blot. The tight junction in the endothelial cells of capillary blood vessels is essential for BBB integrity in the brain [55]. To determine whether STZ-induced hyperpermeability of BBB was due to tight junction alterations, the expression of the tight junction-associated proteins such as ZO-1, occludin, and claudin-5 in the hippocampus was examined at 28 d after drug treatment. As shown in Figure 5A, the levels of ZO-1, occludin, and claudin-5 were markedly decreased in STZ-induced diabetic rats, indicating that hyperglycemia leads to a decrease in BBB integrity by resulting in the loss of tight junction proteins. However, GS-KG9 or GS-E3D treatment significantly decreased the levels of ZO-1, occludin, and claudin-5 as compared with STZ control (Figure 5A,B). These data indicate that GS-KG9 and GS-E3D preserve tight junction integrity by inhibiting the degradation of tight junction molecules and thereby preventing BBB disruption in STZ-induced diabetic rats.

### 3.5. GS-KG9 and GS-E3D Inhibit Macrophage Infiltration in the Hippocampus of STZ-Induced Diabetic Rats

It is known that blood cell infiltration following BBB disruption initiates inflammatory responses, leading to the secondary injury cascade by producing inflammatory mediators. Therefore, we examined the effect of GS-KG9 and GS-E3D on blood cell infiltration by immunofluorescence staining with a macrophage/monocyte cell marker, ED-1 antibody. Immunofluorescence staining showed that ED-1 positive cells were observed in the pyramidal layer and radiatum layer of CA1 at 28 d after drug treatment (Figure 6A, STZ), whereas no ED-1 positive cells were observed in the normal brain (Figure 6A, normal). Furthermore, the number of ED-1 positive cells was significantly decreased in GS-KG9 and GS-E3D-treated groups as compared with the STZ control (STZ, 58 ± 5.8; GS-KG9, 21.8 ± 3.3; GS-E3D, 15.8 ± 3.9) (Figure 6A,B). These results indicate that GS-KG9 and GS-E3D inhibit macrophage infiltration in STZ-induced diabetic rats by preserving BBB integrity.

### 3.6. GS-KG9 and GS-E3D Inhibit Microglia and Astrocyte Activation in the Hippocampus of STZ-Induced Diabetic Rats

A recent report showed that astrocyte and microglia are activated in the hippocampus of diabetic rats and involved in cognitive impairment [56]. Since GS-KG9 and GS-E3D inhibited BBB disruption and macrophage infiltration in the hippocampus of diabetic rats (Figure 3 and Figure 6), we postulated that GS-KG9 and GS-E3D would inhibit microglia and astrocyte activation in the hippocampus of STZ-induced diabetic rats. As shown in Figure 7A, immunostaining with OX-42 antibody, a microglia marker, revealed that microglia morphology displayed a small soma bearing thin-branched or ramified processes in normal rats, indicating a resting state (Figure 7A, normal). However, OX-42-positive cells displayed a significantly-activated morphology, demonstrating cell body hypertrophy and retraction of cytoplasmic processes in STZ-injected diabetic rats (Figure 7A, STZ), whereas GS-KG9 and GS-E3D treatment markedly reduced the number of activated microglia (Figure 7A, STZ+KG9 and STZ+E3D). In addition, immunofluorescence staining with GFAP, an astrocyte marker, showed that astrocytes were also activated morphologically in the hippocampal areas of STZ-treated diabetic rats (Figure 7B, STZ) as compared with normal controls. However, STZ-induced astrocyte activation was markedly attenuated in GS-KG9 and GS-E3D-treated groups (Figure 7B, STZ+ KG9 and STZ+E3D). These results indicate that GS-KG9 and GS-E3D inhibit microglia and astrocyte activation in the hippocampus of STZ-induced diabetic rats.

### 3.7. GS-KG9 and GS-E3D Inhibit the Expression of Pro-Inflammatory Cytokines and Mediators in STZ-Induced Diabetic Rats

It is known that the activation of microglia and macrophages is associated with the production of pro-inflammatory cytokines and mediators. Astrocytes are also active participants in propagating and regulating neuroinflammation [57]. Since our data showed that GS-KG9 and GS-E3D reduced macrophage infiltration and inhibited microglia and astrocyte activation after STZ injection, we expected that GS-KG9 and GS-E3D would inhibit the expression of proinflammatory cytokines and mediators in the hippocampus of diabetic rats. As shown in Figure 8A, the mRNA expression of proinflammatory cytokines such as *tnf-α*, *il-1β*, *il-6* and mediators including *cox-2* and *inos* was markedly increased in STZ-induced diabetic rats as compared with normal controls. Furthermore, their expression was significantly decreased by GS-KG9 and GS-E3D treatment at 28 d after drug treatment (Figure 8A,B). These results suggest that the reduction of neuroinflammation in the hippocampus of GS-KG9 and GS-E3D-treated rats may be mediated in part by inhibiting hyperglycemia-induced glial activation in the hippocampus. However, several reports showed that some inflammatory factors such as IL-6 and Cox-2 are expressed in neurons under the pathological condition [58,59]. Thus, we will examine the effect of GS-KG9 and GS-E3D on the production of proinflammatory factors in neuronal cells.

### 3.8. GS-KG9 and GS-E3D Inhibit Apoptotic Cell Death of Hippocampal Neurons in STZ-Induced Diabetic Rats

There is growing evidence to support the notion that diabetes has adverse effects on the brain, especially on the hippocampus, which is particularly susceptible to neuronal injury and behavior disorder [60,61,62]. Since we showed the inhibitory effect of GS-KG9 and GS-E3D on BBB disruption and neuroinflammation after STZ injection, we postulated that GS-KG9 and GS-E3D would inhibit hippocampal cell damage such as necrosis or apoptosis by STZ-induced hyperglycemia. To examine the cell damage of hippocampal neurons, Nissl staining was performed and hippocampal neurons were counted. As a result, the numbers of pyramidal neurons in hippocampal CA1 region at 32 d after STZ injection were diminished as compared with the normal control group, whereas the loss of hippocampal neurons by STZ injection was attenuated in GS-KG9 and GS-E3D-treated groups (Figure 9A). Quantitative analysis revealed that the number of viable pyramidal neurons in CA1 was decreased to 50% of those of the normal group by STZ, whereas GS-KG9 and GS-E3D treatment significantly attenuated this loss (STZ, 51.4 ± 7.9; GS-KG9, 75.6 ± 6.2; GS-E3D, 78.9 ± 4.7) (Figure 9B). To determine whether the loss of hippocampal neurons in diabetic rats was mediated by apoptotic cell death, TUNEL-staining was performed. As shown in Figure 9C, TUNEL-positive cells were mainly observed in hippocampal CA1 region in STZ-injected rats. However, when compared with the STZ-injected group, TUNEL-positive cells were reduced in GS-KG9 and GS-E3D-treated groups. Quantitative analysis revealed that the number of TUNEL-positive cells in the hippocampal CA1 region was significantly lower in the GS-KG9 and GS-E3D-treated groups than in the STZ-injected control group (STZ, 38 ± 7.9; STZ+GS-KG9, 10.5 ± 3.2; STZ+GS-E3D, 9.8 ± 4.1) (Figure 9C,D). These data imply that GS-KG9 and GS-E3D prevent the loss of hippocampal neurons by apoptotic cell death in STZ-induced diabetic rats.

Neuroinflammation in the hippocampus has been reported as one of the main mechanisms involved in the pathogenesis of hyperglycemia-induced neurodegeneration, which eventually leads to anxiety and depression [8]. Moreover, proinflammatory factors such as IL-6, IL-1β, and TNF-α are known to play a critical role in diabetes-causing anxiety and depression [62,63,64]. Recently, it was reported that anti-inflammatory drugs such as minocycline, taurine, and eugenol reduce blood glucose levels in STZ-induced diabetic rats and inhibit neuropathy and brain damage caused by hyperglycemia [65,66,67,68]. Thus, based on these reports, our results suggest that the neuroprotective effect of GS-KG9 and GS-E3D might be mediated by attenuating BBB disruption, thereby preventing neuroinflammation in the hippocampus in STZ-induced chronic diabetic rats. On the other hand, several studies suggest that oxidative stress is involved in the development of diabetic neurotoxicity and that antioxidant therapy can prevent or reverse hyperglycemia-induced nerve dysfunctions [69,70,71,72]. Recent reports showed that GS-E3D alleviates diabetes-related renal dysfunction by inhibiting the production of 8-hydroxy-2′–deoxyguanosine, a DNA oxidation marker [73]. In addition, E3D and KG9 also showed antioxidant effects in human hair dermal papilla cells proliferation and D-galactosamine-induced liver damage animal model, respectively [74,75]. These evidences taken together suggest that the neuroprotective effect of GS-KG9 and GS-E3D in STZ-induced diabetic rats may be mediated in part by inhibiting oxidative stress. Thus, we will investigate the effect of GS-KG9 and GS-E3D on oxidative stress in a future study. In addition, we will determine the underlying mechanism of the neuroprotective effect of GS-KG9 and GS-E3D in diabetic rats and the effect of these drugs on cognition, anxiety, and depression by chronic hyperglycemia.

## 4. Conclusions

In conclusion, this study provides evidence that bioactive extracts from Korean red and white ginseng (GS-KG9 and GS-E3D) attenuate hyperglycemia and BBB disruption by inhibiting MMP-9 activation in STZ-induced diabetic rats. Furthermore, GS-KG9 and GS-E3D treatment reduced macrophage infiltration and glial activation in the hippocampus, leading to the alleviation of the apoptotic cell death of hippocampal neurons. As a possible mechanism of GS-KG9 and GS-E3D to prevent BBB damage, thereby alleviating neurodegeneration, the inhibitory effect of MMP-9 expression and activation by these drugs can be considered. Thus, we will further investigate the epigenetic regulation by this compound for MMP-9 expression in a future study. Taken together, our results suggest that GS-KG9 and GS-E3D could be used for diabetic patients as a health food and/or therapeutic treatment for preserving the BBB integrity of the brain.

## Figures and Tables

**Figure 1 nutrients-12-02383-f001:**
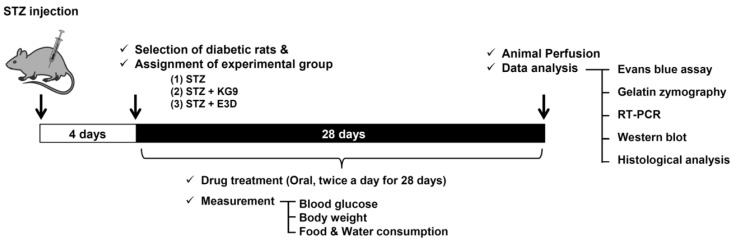
Scheme of experimental design.

**Figure 2 nutrients-12-02383-f002:**
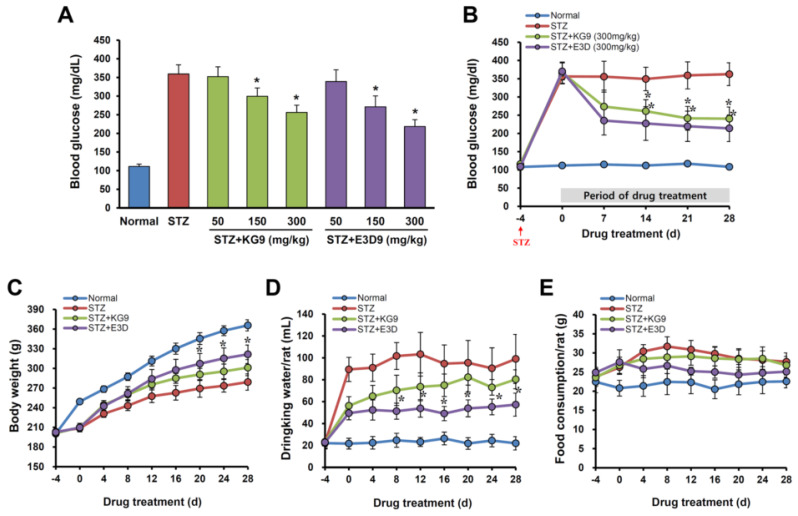
Effects of GS-KG9 and GS-E3D on blood glucose level, body weight, drinking water, and food consumptions in STZ (Streptozotocin)-induced diabetic rats. Diabetes was induced in male SD (Sprague Dawley) rats using streptozotocin (STZ, 60 mg/kg, i.p.). At 4 d after STZ injection, diabetic rats were randomly divided into STZ, GS-KG9, and GS-E3D groups. GS-KG9 and GS-E3D (50, 150, or 300 mg/kg) were administrated orally twice daily for 28 d. (**A**) Concentration of blood glucose at 28 d after drug treatment. (**B**–**E**) The effect of 300 mg/kg of GS-KG9 and GS-E3D on blood glucose level (**B**), body weight (**C**), water consumption (**D**), and food consumption (**E**). Data are expressed as the mean ± SD (*n* = 10 rats/group). * *p* < 0.05 vs. STZ control.

**Figure 3 nutrients-12-02383-f003:**
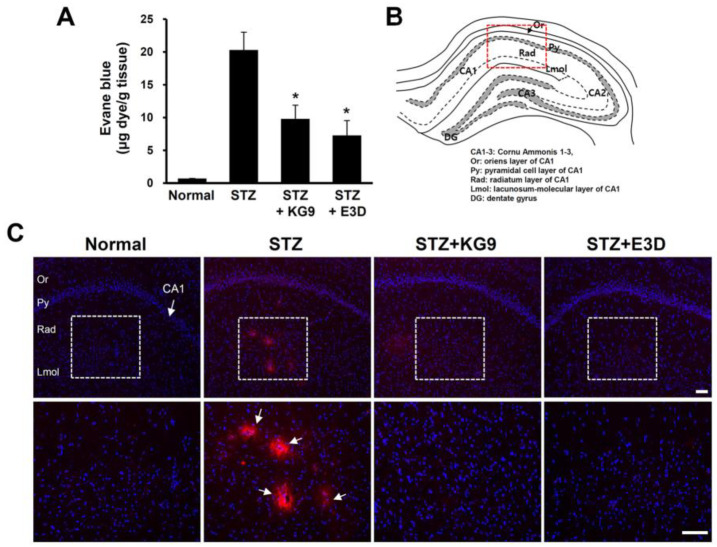
GS-KG9 and GS-E3D inhibit the increase in blood–brain barrier (BBB) permeability in the hippocampus of STZ-induced diabetic rats. For measuring BBB permeability, 5 mL of 2% Evans blue dye was administered i.p. at 28 d after drug treatment, and 3 h later, brain tissue samples were prepared for detection of Evans blue extravasation (*n* = 5 rats/group). (**A**) Quantification of the amount of Evans blue in hippocampus by using fluorometer (excitation at 620 nm/emission at 680 nm). Note that both GS-KG9 and GS-E3D significantly attenuate the increase in Evans blue extravasation in STZ-induced diabetic rats as compared to STZ control. Data are expressed as the mean ± SD. (*n* = 5 rats/group). * *p* < 0.05 vs. STZ control. (**B**) Atlas of hippocampal structure. (**C**) Representative images showing Evans blue extravasation in CA1 area (red box in Figure 3B). The bottom panels are higher magnification images of the box area in the upper panels. Arrows indicate Evans blue extravasation. Red, fluorescence signal; blue, DAPI (4′,6-diamidino-2-phenylindole)stained signal. Note that the fluorescence intensity of Evans blue in the hippocampal radiatum layer (Rad) was significantly reduced in GS-KG9 and GS-E3D (300 mg/kg) treated rats when compared to that in STZ control. Scale bars, 100 μm.

**Figure 4 nutrients-12-02383-f004:**
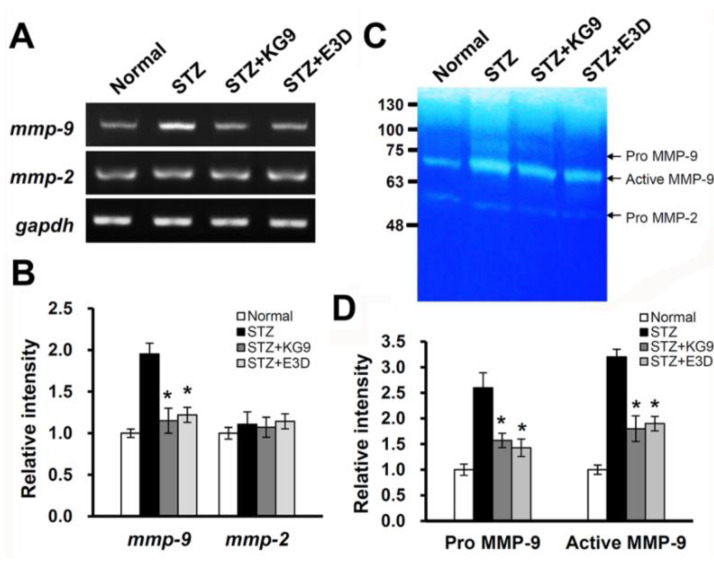
GS-KG9 and GS-E3D inhibit the expression and activity of MMP-9 in the hippocampus of STZ-induced diabetic rats. At 28 d after GS-KG9 and GS-E3D (300 mg/kg) treatment, hippocampal tissue was prepared, as described in the Materials and Methods section. (**A**) RT-PCR of *mmp-2* and *mmp-9*. (**B**) Densitometric analysis of RT-PCR (*n* = 3 rats/group). (**C**) Gelatin zymography. (**D**) Densitometric analysis of zymography (*n* = 3 rats/group). Note that GS-KG9 and GS-E3D significantly inhibit MMP-9 mRNA expression and activation in hyperglycemia-induced rats as compared to STZ control. Data are expressed as the mean ± SD (*n* = 5 rats/group). * *p* < 0.05 vs. STZ control.

**Figure 5 nutrients-12-02383-f005:**
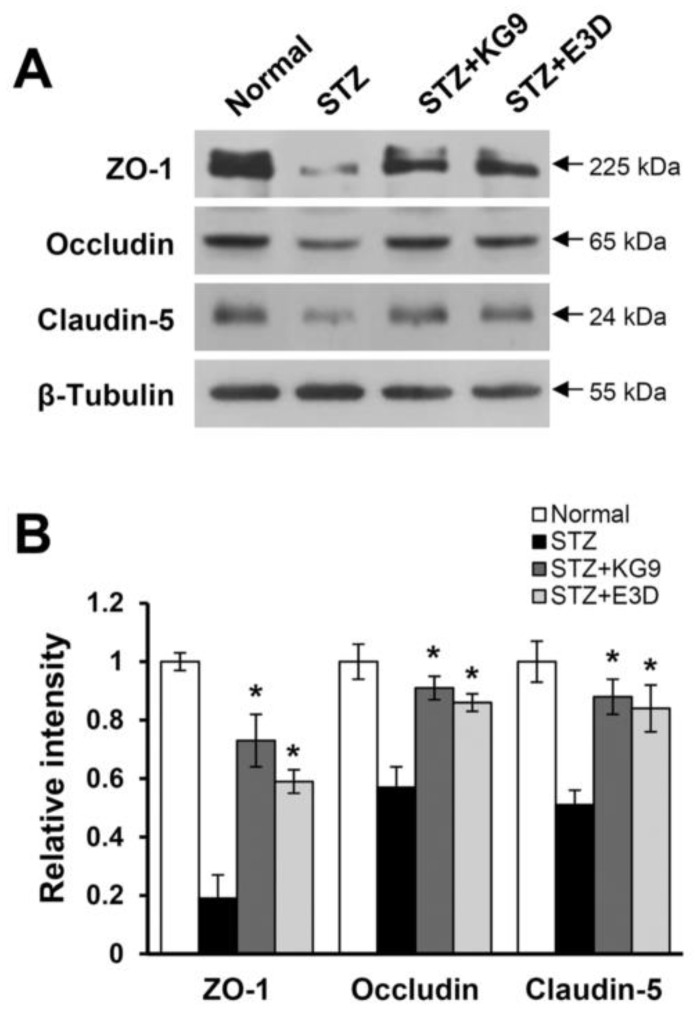
GS-KG9 and GS-E3D alleviate the loss of tight junction proteins in the hippocampus of STZ-induced diabetic rats. Brain tissue and hippocampal extracts from rats treated with GS-KG9 and GS-E3D (300 mg/kg) at 28 d after drug treatment were prepared as described in the Materials and Methods section (*n* = 3 rats/group). (**A**) Western blots for ZO-1, occludin, and claudin-5. (**B**) Quantitative analysis of the Western blots. Data are presented as the mean ± SD. * *p* < 0.05 vs. STZ control.

**Figure 6 nutrients-12-02383-f006:**
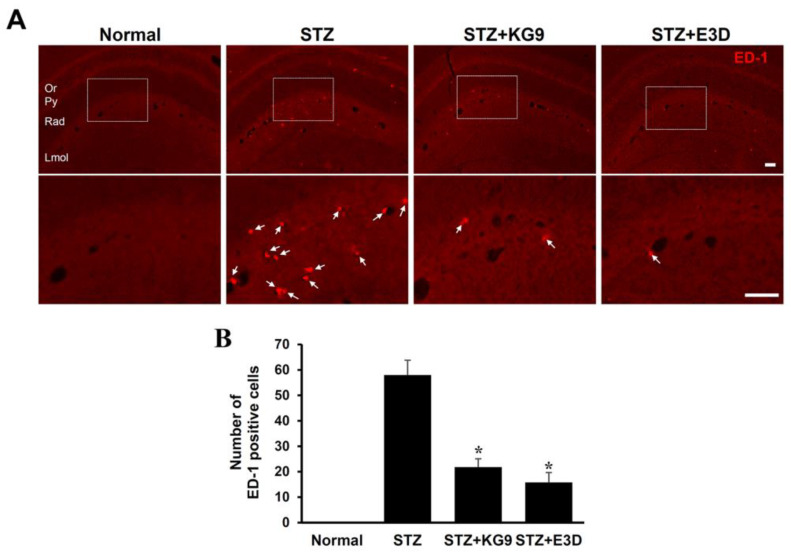
GS-KG9 and GS-E3D inhibit macrophage infiltration in the hippocampus of STZ-induced diabetic rats. Brain tissue from rats treated with GS-KG9 and GS-E3D (300 mg/kg) were prepared at 28 d after drug treatment. (**A**) Representative fluorescence photographs show that ED-1-labeled macrophages in hippocampal CA1 area. The bottom panels are higher magnification images of the box area in the upper panels. Arrows indicate ED-1-positive macrophages. Scale bar, 50 μm. (**B**) Quantification of ED-1 positive cells in CA1. Data are presented as the mean ± SD. * *p* < 0.05 vs. STZ control (*n* = 3 rats/group).

**Figure 7 nutrients-12-02383-f007:**
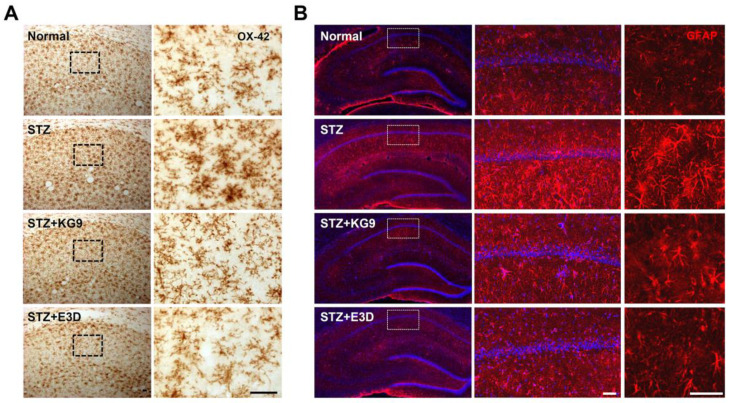
GS-KG9 and GS-E3D inhibit microglia and astrocyte activation in the hippocampus of STZ-induced diabetic rats. At 28 d after GS-KG9 and GS-E3D (300 mg/kg) treatment, brain tissue samples were prepared as described in the Materials and Methods section and we performed immunostaining with microglia marker (OX-42) and astrocyte marker (GFAP). Representative photographs of OX-42 (**A**) and GFAP (**B**) immunohistochemistry showing the hippocampal area. The right panels are higher magnification images of the box area in the left panels. Scale bars, 50 μm.

**Figure 8 nutrients-12-02383-f008:**
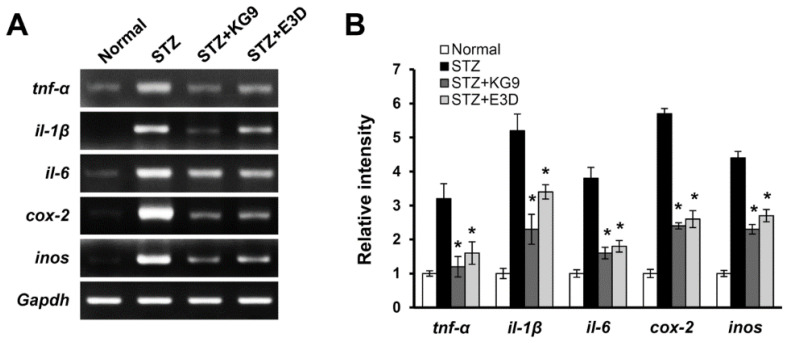
GS-KG9 and GS-E3D inhibit the expression of proinflammatory cytokines and mediators in the hippocampus of STZ-induced diabetic rats. Total RNA from saline, GS-KG9, and GS-E3D (300 mg/kg)-treated hippocampal tissue at 28 d after drug treatment was prepared as described in the Materials and Methods section. (**A**) RT-PCR of *tnf-α*, *il-1β*, *il-6*, *cox-2*, and *inos*. (**B**) Quantitative analysis of RT-PCR. Data are presented as the mean ± SD. * *p* < 0.05 vs. STZ control (*n* = 3 rats/group).

**Figure 9 nutrients-12-02383-f009:**
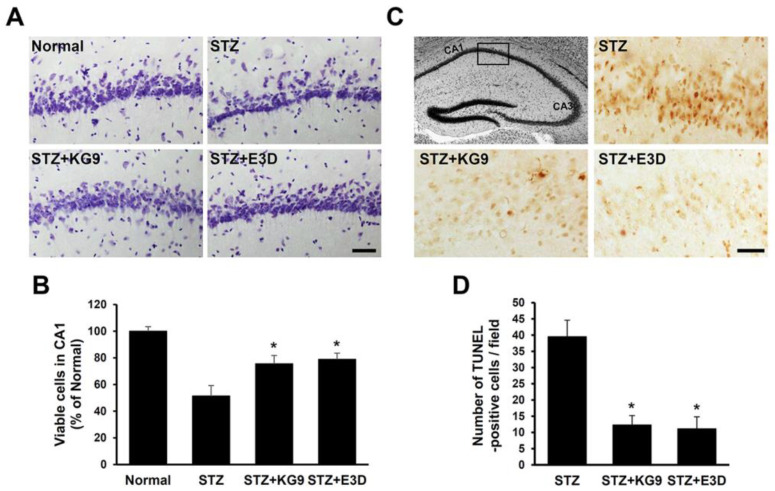
GS-KG9 and GS-E3D inhibit apoptotic cell death of hippocampal neurons in STZ-induced diabetic rats. At 28 d after GS-KG9 and GS-E3D (300 mg/kg) treatment, brain tissue samples were prepared and stained with Cresyl violet and TUNEL (Terminal deoxynucleotidyl transferase dUTP nick end labeling)-stained as described in the Materials and Methods. (**A**) Representative photomicrographs of Nissl staining of the hippocampus CA1 area. Scale bar, 50 μm. (**B**) Quantification of viable cells in CA1 area (*n* = 5 rats/group). (**C**) Representative images of TUNEL staining in CA1 (rectangular box in the brain atlas of left-top panel). Scale bars, 50 μm. (**D**) Quantitative analysis of TUNEL-positive cells (*n* = 5 rats/group). Note that GS-KG9 and GS-E3D significantly decreased the number of TUNEL-positive hippocampal neurons in CA1 area in STZ-induced diabetic rats as compared to STZ control. All data are presented as the mean ± SD. * *p* < 0.05 vs. STZ control.

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
