# Peer review of "Ginseng Extracts, GS-KG9 and GS-E3D, Prevent Blood–Brain Barrier Disruption and Thereby Inhibit Apoptotic Cell Death of Hippocampal Neurons in Streptozotocin-Induced Diabetic Rats"

_nutrients, 2020, doi:10.3390/nu12082383_

Round 1

Reviewer 1 Report

The manuscript by Lee et al. describes effects of ginseng extracts, GS-KG9 and GS-E3D on diabetic rats. In a series of elegant and well-presented experiments, the authors show protective effects of ginseng extracts on physiological parameters of diabetic rats, blood-brain barrier permeability, Mmp9 and Mmp2 activity and tight junction protein expression. In addition, other test like macrophages infiltration, microglia and astrocyte activation, proinflammatory cytokine release and hippocampal neuron apoptosis strengthen the conclusions of the manuscript.

Author Response

Reviewer #1

The manuscript by Lee et al. describes effects of ginseng extracts, GS-KG9 and GS-E3D on diabetic rats. In a series of elegant and well-presented experiments, the authors show protective effects of ginseng extracts on physiological parameters of diabetic rats, blood-brain barrier permeability, Mmp9 and Mmp2 activity and tight junction protein expression. In addition, other test like macrophages infiltration, microglia and astrocyte activation, proinflammatory cytokine release and hippocampal neuron apoptosis strengthen the conclusions of the manuscript.

Response: We really appreciate the Reviewer #1 to review of our manuscript.

Reviewer 2 Report

The aim of the manuscript by Lee et al was to assess whether GS-KG9 and GS-E3D (bioactive ginseng extracts from Korean Ginseng), inhibits BBB disruption followed neuronal death in hippocampus in streptozotocin-induced diabetic rats.

As a general consideration the study is well written, clear and results are interesting. While neuroprotection study is well performed, other points need to be addressed.

Major

The ethical assessment about procedures followed with animals is missing. It is mandatory that Authors ensure that their research complies with the commonly accepted '3Rs' and ARRIVE guidelines, as the politic of Nutrients require. The local IACUC approval is not sufficient.

Chloral hydrate is not acceptable as anaesthetic, as is reported to induce hypnosis and not anaesthesia, it does not provide analgesia, causes marked respiratory depression, and it is also extremely irritating, especially for intraperitoneal use. This is why it has been banned in many Countries for animal research. Finally, the route of its administration is not reported.

The effects of GS-KG9 and GS-E3D per se on healthy rats is missing (see OECD guidelines No. 423). The possibility that the extracts did not produce any signs of toxicity by an oral route should be assessed.

Insulin blood levels should be reported to validate the effects of GS-KG9 and GS-E3D.

Blood glucose level reduction can depend on a downregulation of carbohydrate hydrolysing enzymes in the digestive tract, as often occurs with many natural plant extracts and natural products. As the decreased glucose production from carbohydrate in the gut or glucose absorption from the intestine might depend on α-amylase inhibition, it is mandatory to assess this activity in in vitro assays.

Information about hepatic/blood/brain antioxidant status (for ex. Lipid Peroxidation, CAT and SOD activities) should be reported to validate the effects of GS-KG9 and GS-E3D.

Minor

The dried GS-KG9 and GS-E3D components (line 72-73 and 79-80, respectively) might result unclear to the reader if reported as acronyms. I suggest to briefly explain them or to quote appropriate reference for a better understanding.

GS-KG9 and GS-E3D caused a reduction in blood glucose levels, plateauing at 250 mg/dl. This can still be considered severe diabetes and I suggest tempering the enthusiasm about the effects in the text.

For future experiments I suggest comparing the effects of GS-KG9 and GS-E3D with those of a well-known antidiabetic drug.

Author Response

Response: Please ee the attachment.

Reviewer 3 Report

This paper entitled “Ginseng extracts, GS-KG9 and GS-E3D, inhibit apoptotic cell death of hippocampal neurons by preventing blood-brain barrier disruption in streptozotocin-induced diabetic rats” by Lee et al. described Ginseng’s effects on the diabetic rat’s hippocampus focusing on various factors which associate with BBB and cell death. Data is of interest and experiments are sound. This review has only opinions.

Major opinion

Title: Although this title is impressive, it is not clear enough that Ginseng’s effects on BBB disruption directly resulted in inhibition of neurons’ death. These experiments indicate more likely that Ginseng extracts prevent BBB disruption in streptozotocin-induced diabetic rats and may result in inhibition of neurons’ death.

Minor opinions

  1. 25: It is better to write full spelling of “MMP-9”.
  2. 72: Contents of each ginsenosides of GS-KG9 are not known, not likely GS-E3D (l. 79 and 80)?
  3. 122 and 123: Describe each purpose of antibodies used here to stain what, like in Results and Discussion.
  4. 167: It is better to write full spelling MMP, once, because metalloprotease is not so familiar for general readers.
  5. 317: May be “significantly-activated”.
  6. 339: The authors are sure that all these proinflammatory cytokines produced by only glial cells? At least, it is known that Cox-2 is synthesized in the dentate gyrus neurons under some conditions (for example, ischemia).
  7. 618: Figure C, left top: This image may be Nissl staining. Then, description of Figure C is ambiguous. The present description implies low magnification of TUNEL-staining.

Finally, are there some differences between white ginseng (l. 69) and red ginseng (l. 74)? It seems to be the same spices, Panax ginseng.

Author Response

Response: Please see the attachment.

Round 2

Reviewer 2 Report

The manuscript by Lee et al has been reviewed according to the point raised, and in my opinion is now suitable for publication